# Beyond early initiation: A qualitative study on the challenges of hospital-based postpartum breastfeeding support

**Mai-Lei Woo Kinshella**[1]☯*, **Sangwani Salimu**[2]☯, **Marianne Vidler**[1], **Mwai Banda**[2], **Elizabeth M. Molyneux**[2], **Queen Dube**[3], **David M. Goldfarb**[4], **Kondwani Kawaza**[2,3], **Alinane Linda Nyondo-Mipando**[5]*

**1** Department of Obstetrics and Gynaecology, BC Children's and Women's Hospital and University of British Columbia, Vancouver, Canada, **2** Department of Pediatrics and Child Health, Kamuzu University of Health Sciences, Blantyre, Malawi, **3** Queen Elizabeth Central Hospital, Pediatrics, Blantyre, Malawi, **4** Department of Pathology and Laboratory Medicine, BC Children's and Women's Hospital and University of British Columbia, Vancouver, Canada, **5** School of Global and Public Health, Department of Health Systems and Policy, Kamuzu University of Health Sciences, Blantyre, Malawi

☯ These authors contributed equally to this work.
* Maggie.kinshella@cw.bc.ca (MLWK); lmipando@kuhes.ac.mw (ALNM)

**Data Availability Statement:** All relevant data are within the manuscript and its supporting information files.

## Abstract

Improving breastfeeding practices is key to reducing child mortality globally. Sub-optimal exclusive breastfeeding rates may be associated with inadequate hospital-based postpartum breastfeeding support, particularly in resource-limited health settings such as Malawi. While almost all children in Malawi are breastfed, it is a concern that exclusive breastfeeding rates in Malawi are declining. The objective of this study is to understand postpartum breastfeeding support after delivery at Malawian hospitals from the perspectives of health workers and caregivers. We conducted a secondary analysis of a descriptive qualitative study on health worker and caregiver experiences of breastfeeding support at health facilities in southern Malawi as part of the Innovating for Maternal and Child Health in Africa Initiative. In-depth interviews following a semi-structured topic guide were conducted at three secondary-level district hospitals and one tertiary-level central hospital. Interviews were thematically analysed in NVivo 12 software (QSR International, Melbourne, Australia). We interviewed a total of 61 participants, including 30 caregivers and 31 health care workers. Participants shared the following themes: 1) a focus on early initiation of breastfeeding, 2) inadequate follow-up on breastfeeding practice, and 3) feasibility challenges and local solutions. There was an emphasis on early initiation of breastfeeding, which was challenged by maternal exhaustion after delivery. Study participants reported poor follow-up on breastfeeding practice after initial counselling and reacting to adverse outcomes in lieu of adequate monitoring, with a reliance on caregivers to follow-up on challenges. There was poor support for facility-based breastfeeding after initial counselling post-delivery, which revealed an overall neglect in hospital postpartum care for those considered to be in good health after initial assessment. We recommend the development of indicators to track continued facility-based breastfeeding, identify vulnerable infants at-risk of feeding problems,

**Funding:** MWK, ALNM, KK, QD and DG were funded by the Canadian International Development Research Centre (IDRC), Global Affairs Canada (GAC) and the Canadian Institutes for Health Research (CIHR). Project ID is 108030. The funders had no role in study design, data collection and analysis, decision to publish, or preparation of the manuscript.

**Competing interests:** I have read the journal's policy and the authors of this manuscript have the following competing interests: ALNM is an editorial board member of the journal. All other authors have declared that no competing interests exist.

and strengthening care in postnatal wards, which is currently as neglected component of maternity care.

## Introduction

The World Health Organization (WHO) recommends early initiation of breastfeeding within an hour after birth and exclusive breastfeeding for 6 months, followed by complementary feeding up to two years [1]. Exclusive breastfeeding is defined as the infant receiving only breastmilk, in contrast to partial breastfeeding where other liquids and/or solids are also provided [2]. In comparison to exclusive breastfeeding, partial breastfeeding (RR 1.83, 95% 1.45–2.32) and no breastfeeding (RR 10.88, 95% CI: 8.27–14.31) is associated with significantly high risk of neonatal death [2]. Following the WHO recommendations, increasing optimal breastfeeding practices to near universal levels are estimated to prevent over 800,000 child deaths each year in low- and middle-income countries (LMICs), corresponding to 13.8% of deaths under two years old globally [3]. As key to reducing child mortality, improving early health and nutrition, with life-long impacts on long-term health, cognitive development and participation in society, optimal breastfeeding practices play an important role in achieving Sustainable Development Goals to end poverty and fight global inequality [4–6].

However, breastfeeding practices remain sub-optimal, with only 43% of newborns being breastfed within an hour of birth and 44% of infants under six months of age being exclusively breastfed [1, 7]. Though breastfeeding rates are higher in LMICs in comparison to high-income countries (HICs), LMICs carry a disproportionate burden of child deaths globally and breastfeeding trends remain sub-optimal in LMICs [2]. There may also be inconsistencies between early initiation and exclusive breastfeeding rates. While Malawi exceeded the global 2030 early initiation target rate of 70%, data from 2015 reported a shortfall between the early initiation rate (76%) and the rate of exclusive breastfeeding (61%) [8]. This highlights that even in countries with high early initiation to breastfeeding, continued optimal breastfeeding practices remain a challenge.

Breastfeeding practice has been shown to improve when support is given to mothers in the facility, community, or family [9–11]. A Cochrane review found that offering breastfeeding support to women was associated with a 12% reduced risk of stopping exclusive breastfeeding before six months of age (RR 0.88, 95% CI 0.85–0.92, 46 studies) [10]. A review found that baby-friendly hospital support was the most effective intervention to improve breastfeeding rates, associated with a 49% increase in exclusive breastfeeding (RR 1.49, 95% CI 1.33–1.68) and a 20% increase in early breastfeeding initiation (RR 1.20, 95% CI 1.11–1.28) [11]. While almost all health facilities reported routinely providing breastfeeding counselling during antenatal care (ANC) in Malawi, only 40% of ANC providers reported recent training in breastfeeding and less than 40% reported postnatal care (PNC) for mother or infant within the first hour of delivery [12, 13]. Findings from the Malawian study highlight low rates of PNC and a need to support ANC and PNC provider training for improved breastfeeding practices.

While cross-sectional studies in Malawi confirm high rates of early initiation of breastfeeding from 77% to 95% [14, 15] and almost all children (98%) have been breastfed at some time [8], it is of concern that the rate of exclusive breastfeeding is decreasing. Between 2010 and 2016, the percentage of infants who were exclusively breastfed in Malawi declined by 11% [8]. There is a gap in supporting mothers to continue breastfeeding exclusively. The purpose of this study is to explore the perspectives and experiences of health workers and caregivers on the availability and content of postpartum breastfeeding support after delivery at Malawian hospitals.

## Methodology

### Study design and setting

This is a secondary analysis of a descriptive qualitative study on health worker and caregiver experiences with breastfeeding support at four health facilities in southern Malawi. The objective of the primary study was to identify overall gaps to be addressed in order to improve breastfeeding support in Malawi [16], while the objective of the secondary analysis was to better understand the availability and content of postpartum breastfeeding support with attention to early initiation of breastfeeding compared to post-initiation counselling. In-depth interviews to elicit rich data to better understand the social phenomenon were conducted [17]. Part of the Innovating for Maternal and Child Health in Africa (IMCHA) initiative, this study is embedded in the "Integrating a neonatal healthcare package for Malawi" project, which seeks to inform the scale-up of low-cost and locally appropriate innovations to improve newborn care at low-resource health facilities. The study is reported based on the "Consolidated criteria for reporting qualitative research" (COREQ; **S1 Table**) [18].

We conducted the study at four hospitals in southern Malawi, one a tertiary-level facility and three secondary-level facilities serving rural populations in their districts. The tertiary hospital and two district hospitals are publicly owned while the third district hospital is a private non-profit facility owned by the Christian Health Association of Malawi. Though privately owned, the mission hospital has a service agreement with the government to provide maternal and child health services free to patients. The decision to choose the three district facilities was guided by the Ministry of Health to represent a variety of health management structures available in Malawi [19]. For facility assessments that further describe these district facilities and their quality of neonatal services provided, see Kawaza et al. 2020 [19].

### Recruitment and selection

We employed purposive sampling to recruit health workers engaged in decision-making and neonatal care, breastfeeding mothers and their relatives. We purposively recruited nurses and clinicians working in neonatal units, nurses in charge of the ward, registrars and pediatric consultants at the tertiary facility plus district health officers, district medical officers, district nursing officers and nurses in charge of the pediatric ward at district facilities. Study staff approached health workers in person and/or by phone and asked for an interview after briefing them about the study. Mothers and family members involved in providing care to the infant were approached in the postnatal ward, neonatal unit or at visiting area outside the ward. We prioritized male partners and grandmothers as they are influential sources of information and support for new mothers in Malawi [20]. Based on the limited number of staff available for neonatal care, particularly at the district hospitals, a sample size of 5–10 health workers and 5–10 family members at each site were estimated to reach data saturation with a variety of perspectives. All health workers we approached agreed to participate while three caregivers declined: a mother cited unpreparedness, a grandmother felt she did not have anything to share and a male partner cited inadequate time. No repeat interviews were conducted.

### Data collection

We used a semi-structured interview approach to allow participants to elaborate on their personal experiences [17]. A team of five female and one male data collectors conducted face-to-face interviews overseen by ALNM (PhD in Health Systems and Policy) and SS (MPH). Of the data collectors, four are nurses and two are social scientists. Data collectors were hired as part of the project, trained in qualitative research methods by ALNM and MWK (MA in Medical

Anthropology) and did not know participants prior to the study. Data was collected between April and June 2020. The interview guides were developed for this study (**S1 and S2 Texts**). Pilot interviews were conducted at the tertiary facility to refine the tools. Because of the exploratory nature of the study, data from pilot interviews were included in the analysis. After providing consent, interviews were conducted in secluded settings within the hospitals. Health worker interviews were approximately 50–60 minutes in length while caregivers' interviews were approximately 30–45 minutes. The broad questions that guided the interviews were as follows:

1. How do health workers support mothers after the first time they started breastfeeding?

2. What other support would caregivers want from health workers in order to ensure that the baby keeps breastfeeding?

We probed further after the initial broad questions to capture more depth on the subject as per topic guide. Interviews were conducted in either English or Chichewa, according to participants' preferences. All interviews were recorded with permission, using a digital audio recorder and each recorded interview was assigned a unique identification number. Audio records and completed transcripts were stored on a password-protected computer. Quality of data was ensured by making field notes during the interview and applying member checking at the end of each section within the interview [21, 22]. No participants opted to review their transcripts.

### Data analysis

Data were managed using NVivo 12 software (QSR International, Melbourne, Australia). We applied the thematic approach as developed by Braun and Clarke for analysis [23]. Preliminary analysis commenced during the data collection period through debriefs to discuss emerging issues with the data collection team and continued as audio recordings were transcribed verbatim, with any portions in Chichewa simultaneously translated into English. Researchers (SS, ALNM, MWK) listened to the audio tapes and read the transcripts several times to familiarize themselves with the data to develop the coding framework. During the primary analysis, SS coded the interviews using the framework with a second round of coding by MWK and ALNM for quality control. Discrepancies were discussed between coders until consensus was achieved. A secondary analysis was coded by MWK to specifically elucidate themes around initiation and post-initiation support (**Fig 1**). Coded transcripts and summaries of emergent themes were reviewed with SS and ALNM.

### Ethics statement

This project received ethics approval from the University of Malawi, College of Medicine (P.08/15/1783) and the University of British Columbia (H15-01463-A003). Formal written consent was obtained from all study participants.

### Results

We interviewed 61 participants including 30 caregivers and 31 health care workers (**Table 1**). Of the 30 caregivers, there were 17 mothers, 10 fathers, two grandmothers, and one aunt. The median age for mothers was 27 years old and 32 years for fathers. Mothers had a median of 7.5 years of schooling completed and 6 years for fathers. Of the 31 health workers, there were 17 nurses, two nursing officers, seven clinical officers, three district health management officials (district health officer, district medical officer, district nursing officer), and two pediatric specialists. Healthcare workers had a median 5 years of experience and 52% (16 of 31) received

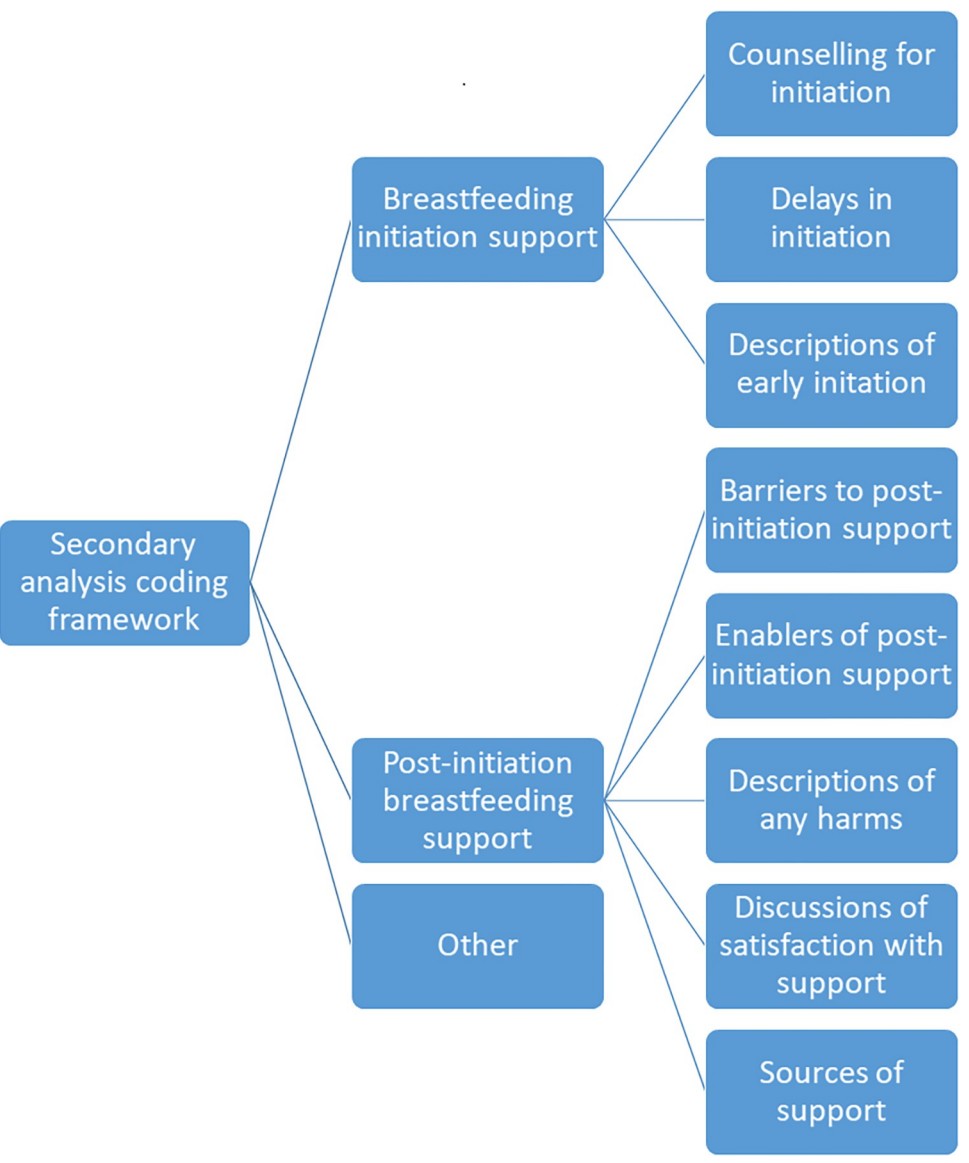

**Fig 1. Coding framework for secondary analysis.**

formal training for breastfeeding support. There were 19 participants from the tertiary hospital and 42 from district hospitals.

Participants shared three major thematic areas for postpartum breastfeeding support at Malawian hospitals: 1) early initiation focused, 2) inadequate follow-up on breastfeeding practice, and 3) feasibility challenges and local solutions. Themes and sub-themes are illustrated in **Fig 2** and described further below. See also **S2 Table**.

## Early initiation focused

**"It did not take long after the baby was born**. . ."**–Counselling for early initiation of breastfeeding.**   Health workers across cadres were well aware of early initiation of breastfeeding and highlighted its importance. There was a consistent message of initiating breastfeeding as soon as the mother and infant were stable, ideally within 30 minutes to an hour after

**Table 1. Participant characteristics.**

|  | Number of participants (%) |
|---|---|
| HEALTHCARE WORKERS (n = 31) |  |
| Profession |  |
| Nurse | 17 (55%) |
| Nurse supervisor | 2 (6%) |
| Clinical officer | 7 (23%) |
| District health management* | 3 (10%) |
| Pediatric doctors | 2 (6%) |
| Location |  |
| Rural district hospital | 24 (77%) |
| Urban tertiary hospital | 7 (23%) |
| Age (median [IQR]) | 32 years old [29, 41.5] |
| Years of experience as a health care worker (median [IQR]) | 5 years [3, 6.8] |
| Years of experience working with newborns (median [IQR]) | 3 years [2.1, 5] |
| Trained to offer breastfeeding support | 16 (52%) |
| CAREGIVERS (n = 30) |  |
| Relationship to the newborn |  |
| Mother | 17 (57%) |
| Father | 10 (33%) |
| Grandmother | 2 (7%) |
| Aunt | 1 (3%) |
| Location |  |
| Rural district hospital | 18 (60%) |
| Urban tertiary hospital | 12 (40%) |
| Age (median [IQR]) |  |
| Overall | 29 years old [26, 34.3] |
| Mothers | 27 years old [23.8, 29.5] |
| Fathers | 32 years old [28, 39] |
| Years of education† (median [IQR]) |  |
| Overall | 7 years [5, 9.5] |
| Mothers | 7.5 years [6, 8] |
| Fathers | 6 years [4, 10] |

IQR–interquartile range

*District health management officials include the district health officer, district medical officer, district nursing officer

† 8 years to complete primary school

delivery. Participants shared that early initiation was promoted so the infant benefits from receiving colostrum, the first milk produced immediately after delivery.

> "Usually when women are coming out from labor and we have seen that they are just fine. . .they start breastfeeding right away. . . 30 minutes after delivery, the baby must be found being breastfed. The reason is [that] the first milk that comes out is very important because it contains colostrum and it has some antibodies so that that the baby should not fall sick frequently." *Tertiary hospital nurse*

Interviews from mothers and her relatives confirmed the emphasis on early initiation. Caregivers reported that they were counselled to start breastfeeding soon after delivery.

**Fig 2. Postpartum breastfeeding support at Malawian hospitals themes and sub-themes.**

"It did not take long after the baby was born. . . Just after the baby was born, it only took approximately fifteen minutes then they told me that the baby is supposed to exclusively breastfeed because when doing so, the uterus goes back to its position and it also helps the baby to be healthy." *Mother from a district hospital*

**"And they forget**. . .**"–Issues around breastfeeding counselling after delivery.** The process of preparing a new mother to breastfeed included cleaning the mother up after delivery, HIV counselling for seropositive mothers, and counselling mothers to breastfeeding. This happened in the labour ward by midwives who facilitated the delivery. Counselling messages varied between health workers but often included information on the benefits and frequency of breastfeeding. Some health workers also mentioned counselling mothers on correct positioning, attachment, and hygiene. However, when asked by researchers in our study, mothers often recalled only basic messages of the counselling they received after delivery. Health workers also highlighted that mothers sometimes forgot the counselling they taught them.

"They said we should breastfeed exclusively. . . [and] we should take care of the baby. . .that's all that l remember" *Mother from a district hospital*

"I couldn't call it refusing. . . You provide the information, they practice and everything, and they forget. . . I have never had an example of refusing someone who does not want to do this. But sometimes after intensive teaching, after intensive support, then you go there you find them back to the old ways of doing things they are used to." *District hospital nurse*

## Inadequate follow-up on breastfeeding practice

**"You teach her, you leave her**. . .**"–Poor follow-up on breastfeeding practice after postnatal counselling.** Health workers highlighted a short two hour window of direct monitoring after delivery when the mother and infant were still in the labour ward before moving to the postnatal ward for recovery or another ward for further care in the case of complications. The average time to discharge after delivery for mothers and infants without complications was between 12–24 hours. Monitoring for breastfeeding practices lacked the urgency of health emergencies in the labour ward, such as maternal hemorrhage and neonatal resuscitation. While counselling and guidance to initiate may occur after delivery, there was a lack of routine monitoring and follow-up lacked observing breastfeeding practice.

"What is normally focused on is: how is the mother? The mother is not bleeding; the mother is fine. How is the baby? The baby is fine [and] is breastfeeding well. Fine, the mother can go to the postnatal ward. . .uhh. . . We have left the mother who is not bleeding, who is fine, the vitals are fine and is breastfeeding and you just take it for granted that everything is alright but we don't know this. Is it (breastfeeding) really happening? Is the support the right one? Is it being done the right way? . . .[P]eople didn't even bother even to go there (to postnatal) to do monitoring. . ." *District hospital nurse*

"The moment we do the health talk, we assume that every mother has understood. Unless she comes to ask for. . . individual assistance. We never go there to say, let me see how you are feeding the child, is your child feeding?" *Tertiary hospital nurse-midwife*

"From my observation, it is not monitored except for the two hours we had we have with the mother in labor. . . [then] you refer them to the recovery room (postnatal ward). So I

would say the only time that we do the observations are these two hours. . . if you are in short [staffed] the moment, you finish with the mother, you give her the baby to breastfeed. . . you teach her, you leave her. . .you are moving that mother to the recovery so to me, I would say monitoring is not done. . .” *Tertiary hospital nurse-midwife*

**"So we were just managing symptoms**. . .**"–Reacting to adverse outcomes in lieu of adequate monitoring.** Some health workers shared that beyond early initiation to breastfeeding, a key challenge was understanding if infants were breastfeeding enough and emphasized that poor positioning and attachment may affect the effectiveness of breastfeeding practice. Poor breastfeeding practices increased the risk of adverse infant outcomes, which some health workers highlighted was remedied after teaching mothers to properly breastfeed. Due to a lack of regular monitoring of postnatal mothers and infants, health workers shared that they often had to react to adverse health outcomes rather than monitoring vital signs that would indicate emerging complications before they become severe, which may come too late for some as the narratives below reveal.

"[W]hen we expressed the milk on the mother and the baby breastfed and after three meals we saw the baby changing (improving). . . That time, we had only the nursery ward and we were not monitoring glucose level of these babies so we were just managing symptoms" *District health officer*

"Once we were on a night shift. . .she was a prim (first-time mother) and postnatal. . .. This one was breastfeeding the baby while she is asleep. . .so [she] was told that that is bad and that is not the way to do it, because the baby may choke on the milk. . .. So, it happened that, it was around AM hours (middle of the night), the guardian (female relative) knocked and said come and see the baby. When we went there. . .I think the baby had aspirated. So, what was coming out was milk mixed with blood from the nose and the mouth. Then, we rushed the baby to the nursery. . .but it didn't do us any good. That's how the baby died. Now, the guardian started accusing the child (the mother) . . . everybody in the morning was accusing her, that she didn't follow what she was told, a lot of things. But at the same time, we had to understand that maybe she (the mother) wasn't feeling alright. . .. I would say I think, us as midwives, if we did our jobs like during the night. . . [If] we didn't rest as in waiting for people to come and tell us their problems what they are facing, rather we go in doing ward rounds, you know. . . walk around and see how people are doing, maybe we could have seen that this mother was not feeding the baby the right way and we could have saved the baby." *Tertiary hospital nurse-midwife*

## Feasibility challenges and local solutions

**"One just does what is more important** . . .**"–Staffing shortages and prioritization.** A reoccurring theme shared by health workers was human resource constraints that led to prioritizing emergencies in maternity rather than follow-up with mothers and infants assumed to be in stable condition. Health workers shared about numerous deliveries occurring daily at each of their health facilities, which posed challenges to adequately monitor vital signs and observe breastfeeding in practice for all patients, particularly at night when there was less staff on duty. Staff from the tertiary hospital reported 120 postnatal beds divided between six nurses to cover. At the district hospitals, staff reported on average 20 or more deliveries daily, covered by two nurses on duty during the day and one nurse during the night.

"It's a challenge because we are understaffed. . . so most of the time it is a problem to monitor lactation unless if we are assessing those with challenges. . .. Ideally, soon after mothers are admitted in the ward, they are supposed to be monitored on how they are holding the babies to the breast and if they are producing milk, and on consecutive assessments, we have to assess if they are producing milk and if they are holding the baby correctly to the breast. . . During the day, we have six nurses and three during the night and they are two postnatal wards, 120 beds in total. . . so this poses as a challenge to assess how the women are breastfeeding their babies" *Tertiary hospital nursing officer*

"Most of the time, we fail to monitor effectively because of a shortage of staff. We are very few which makes us just to monitor those cases that we believe are critical. . . For example, it happened to me, on that day, I had a patient who was bleeding and the other patient had convulsions. The situation made me be in dilemma and choose to leave women who were breastfeeding and attend those cases." *Tertiary hospital nurse*

"The health workers feel they are busy people. . .If I have helped a mother deliver, we are done and we move to another patient, and this makes us not to focus on following the mother. . .." *District medical officer*

**"You train the mother**. . .**"–Reliance on caregivers to monitor for breastfeeding challenges.** Health workers reported coping with demands on their attention by relying on caregivers to monitor and seek help when challenges arise. Responsibility for care was conceptualized to be the role of the mother and her accompanying relatives, which was shared by both health workers and caregivers alike.

"The idea is not that the nurse should be there all the time, so you train the mother. You empower the mother. You teach them and observe if they are following the norms. . ." *Tertiary hospital consultant*

"When mothers from the C-section (caesarean section delivery) are admitted in the postnatal ward and have started breastfeeding the baby, we rely on the guardians to check the mothers on how well they are breastfeeding their babies because we cannot assess the patients now and then to see if they are following the things they were advised to do." *Tertiary hospital nursing officer*

"If the baby is not breastfeeding, as a mother I have to notify the doctor about my situation. So they look for a way to help the child for it to start breastfeeding" *Mother at a district hospital*

**"My roles are many"–The benefits and challenges of caregiver support.** Caregivers and health workers frequently described female relatives accompanying mothers as playing a vital role in supporting facility-based breastfeeding. Health workers appeared to be aware that mothers may be tired and may have challenges adhering to or remembering recommendations, so her relatives were also engaged to support.

"They explained that I should also make sure that baby is feeding because sometimes the mother gets tired and falls asleep" *Grandmother a district hospital*

"They told us that the babies should be exclusively breastfed. . . Since she (the mother) is not well, they (health workers) are explaining everything to me. . .They told us that the babies should be in a good position when breastfeeding." *Grandmother a district hospital*

"Most of the times, [the] caregivers here are the grandmothers who usually help the patient by taking the breast and feed the baby. They teach such mothers how to breastfeed. They also encourage the woman to keep on breastfeeding." *Tertiary hospital nurse*

However, caregivers may also reduce breastfeeding opportunities. As reported by both care-givers and health workers, grandmothers in particular tended to take the baby away from the mother to care for the infant and to give the mother space to rest. However, the physical sepa-ration of the mother and infant reduced the ability to breastfeed, which was harmful to the child in the long term, especially low birthweight infants.

"You try to give support to help the mother and everything, then you leave [and] the guard-ian, especially the elderly, come in [and] get the baby. . . The babies usually don't stay with their mothers. We push them for the babies [to stay] with their mothers [but] then the guardians think they love the patient (the baby) very much and they get the baby again for a long time. You try to push them to say, "No, no, this baby has to be staying with the mother almost all the time to help them when you leave." They say, "No, our patient is so sick". . . .yes, something like that, the baby is usually in the hands of guardians." *District hospital nurse*

## Discussion

Our research with health workers and caregivers at Malawian hospitals on facility-based breastfeeding support after delivery revealed an emphasis on early initiation of breastfeeding but a gap in continued medical staff support. Health worker monitoring, correction of inap-propriate breastfeeding practices and personalized support was rare after the mother and infant moved from the labour ward into the postnatal ward. In the postnatal ward, follow-up that the baby was breastfeeding and emergence of adverse complications relied on self-report by caregivers.

While several caregivers and health workers in our study described providing and receiving postnatal breastfeeding counselling immediately after delivery, participants also highlighted challenges in how much of the information was retained. One issue may be that mothers are tired after delivery and unable to retain information. A study of 160 postpartum women in the United States found the more exhausted a mother reported feeling, the more she also reported frustrations and difficulties in breastfeeding [24]. In the same study, nursing support to pro-mote a relaxed state was significantly associated with increased effective breastfeeding practice as reported by mothers on discharge from the hospital [24]. Likewise, a study with 374 post-partum women in Turkey found that maternal tiredness was associated with delayed early ini-tiation and higher rates of providing prelacteal feeds [25]. Another challenge may be low maternal education rates, particularly as three of the four research sites were in rural hospitals where 86% of women may not have completed primary education [8]. A meta-analysis of 36 studies found that women with high education attainment were over two times more likely to initiate breastfeeding early in comparison with lowest maternal education levels (RR 2.28, 95% CI: 1.92–2.70) [26]. Clinical explanations of technical details such as attachment and position-ing may be difficult to understand and remember, particularly if mothers are fatigued in the early postpartum period.

Our study revealed that while support of a mother's relatives was invaluable, they may have different ideas on breastfeeding and postpartum care. In particular, grandmothers may encourage alternative practices and mothers may not have decision-making power [16]. This builds on findings of a systematic review that found mothers were 12% more likely to initiate

breastfeeding if grandmothers had a positive attitude, yet if grandmothers had negative attitudes towards breastfeeding, there was an up to 70% decrease in the likelihood of breastfeeding [27]. Previous research in Malawi has shown that grandmothers are highly influential as household decision-makers but are rarely involved in health education and their perspectives are not often taken into consideration when engaged [20]. There was some indication of competing concepts of care as health workers described how grandmothers devoted time to infants to show love and allow mothers to rest, but reduced opportunities for breastfeeding. Though family support was helpful given limited staffing, family dynamics also need to be considered, particularly with young mothers.

A stark gap emerged in our research on the follow-up of postpartum mothers at health facilities. A facility assessment of the quality of newborn care available in rural district hospitals in Malawi found that hospitals largely met current standards of care for early and exclusive breastfeeding with little improvements needed, but there were no questions on supporting the maintenance of breastfeeding practice beyond early initiation [16]. This speaks more broadly to a gap in postpartum care in general. Health workers in our study spoke about the need to focus on the next delivery, and follow-up of mothers and their infants in the postpartum ward was neglected. Strengthening health systems with sufficient staff, equipment, and management structures is critical as facility births in sub-Saharan Africa have increased dramatically by 85% (1990 to 2010 Demographic and Health Surveys) [28] with an increase from 55% to 91% of births in Malawi from 2000 to 2016 [8, 29]. Investing in the capacities of hospitals to support breastfeeding initiation and maintenance helps to strengthen quality postpartum care more broadly.

Exclusive breastfeeding could be increased through enhancing and investing in health facility capacity to support breastfeeding with continuity from antenatal care, counselling for early initiation, follow-up until hospital discharge and postpartum care visits [30]. This would include revamping, expanding and institutionalizing the baby-friendly hospital initiative through more practical training of staff in breastfeeding protection, promotion and overall postpartum support [30, 31]. This training needs to emphasize continuity between the antenatal and postnatal period, with attention to vulnerable populations like adolescents, primigravidas, those who delivered through caesarean section, family-centred care with involvement of husbands/partners and birth companions, and integration with programmes for the prevention of mother-to-child transmission of HIV [16, 30, 31]. Staff capacity could be supported by task-shifting of the roles of education and supporting mothers with breastfeeding to lay health workers [16]. There is evidence that a combination of professional support and peer support by trained and experienced lay health workers supports the continuation of breastfeeding, with the role of peer support identified to be the most important during the postnatal period [32, 33]. However, there is a gap in research on implementing breastfeeding peer support programs in LMICs, where health facilities may be resource-constrained and staffing capacity could be greatly bolstered by such programs [32, 34]. Additionally, strengthening staff capacity at the health facility can be accompanied by providing community-based strategies to support exclusive breastfeeding and implementing communication campaigns tailored to the local context [35]. Malawi is resource-constrained and had halted recruitment of health care workers, which has prolonged the staff shortages. There are plans to revise the postpartum care under pipelines and this study will feed into those plans.

## Limitations

A limitation of this study is that it is a secondary analysis of data and relied on self-reporting of postpartum counselling messages and follow up from health workers and caregivers. Our

researchers did not observe counselling sessions received by caregivers interviewed in this study. More research can further explore the content and quality of breastfeeding counselling messages to elucidate beyond that it happened, and to understand how counselling was facilitated. However, it is worth noting that we systematically defined a priori, the inclusion and exclusion criteria to identify potential study participants, and recruitment involved the selection of a diverse sample with regards to socio-demographic characteristics of women and guardians. While we acknowledge that the findings of this study cannot be generalized to the entire population, we argue that it can be applied to populations accessing care at other Malawian public facilities. Facility assessments in our study district hospitals had similar results to a wider assessment using the same tool in five districts across Malawi suggesting generalizability of these challenges in neonatal and postnatal care [19]. A strength of this study is that it was conducted with neonates still in the hospital hence reducing the effects of recall bias among mothers and guardians. The use of multiple sources also helped to validate the findings.

## Conclusion

Our secondary analysis highlights the importance of postpartum breastfeeding support at health facilities in Malawi. Caregivers and health workers interviewed in our study shed light on the discrepancy between a higher rate of early initiation compared to rates of exclusive breastfeeding in Malawi, particularly highlighting the gap in continued facility-based support after initial counselling post-delivery. Poor breastfeeding support at facilities after initiation reveals overall neglect in postpartum care for those considered in stable health after an initial assessment. We recommend the development of better indicators and tracking of continued breastfeeding support at health facilities, with further research and quality improvement in postnatal recovery wards to strengthen this neglected component of maternity care.

## Supporting information

**S1 Table. COREQ checklist.**
(PDF)

**S2 Table. Summary of qualitative results.**
(PDF)

**S1 Text. Interview guide for health workers.**
(PDF)

**S2 Text. Interview guide for caregivers.**
(PDF)

## Acknowledgments

This manuscript is part of the "Integrating a neonatal healthcare package for Malawi" project within the Innovating for Maternal and Child Health in Africa (IMCHA) initiative. The authors are grateful to the IMCHA team for their support, the study participants for their voluntary participation, and the Directors of the various institutions included in the study for allowing us to conduct the study in their facilities.

## Author Contributions

**Conceptualization:** Mai-Lei Woo Kinshella, Elizabeth M. Molyneux, Queen Dube, David M. Goldfarb, Kondwani Kawaza, Alinane Linda Nyondo-Mipando.

**Data curation:** Mai-Lei Woo Kinshella, Alinane Linda Nyondo-Mipando.

**Formal analysis:** Mai-Lei Woo Kinshella, Sangwani Salimu, Alinane Linda Nyondo-Mipando.

**Funding acquisition:** Queen Dube, David M. Goldfarb, Kondwani Kawaza.

**Investigation:** Mai-Lei Woo Kinshella, Sangwani Salimu, Alinane Linda Nyondo-Mipando.

**Methodology:** Mai-Lei Woo Kinshella, Alinane Linda Nyondo-Mipando.

**Project administration:** Sangwani Salimu, Mwai Banda.

**Supervision:** Mai-Lei Woo Kinshella, Marianne Vidler, Elizabeth M. Molyneux, Queen Dube, David M. Goldfarb, Kondwani Kawaza, Alinane Linda Nyondo-Mipando.

**Writing – original draft:** Mai-Lei Woo Kinshella, Sangwani Salimu.

**Writing – review & editing:** Mai-Lei Woo Kinshella, Sangwani Salimu, Marianne Vidler, Mwai Banda, Elizabeth M. Molyneux, Queen Dube, David M. Goldfarb, Kondwani Kawaza, Alinane Linda Nyondo-Mipando.

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
