## [Decision Letter · Decision Letter 0]

20 Jun 2022

PGPH-D-22-00362

“You just take it for granted that everything is alright”: Beyond early initiation to challenges of continued facility-based breastfeeding support in Malawi

Dear Dr. Kinshella,

Thank you for submitting your manuscript to PLOS Global Public Health. After careful consideration, we feel that it has merit but does not fully meet PLOS Global Public Health’s publication criteria as it currently stands. Therefore, we invite you to submit a revised version of the manuscript that addresses the points raised during the review process.

EDITOR: Please insert comments here and delete this placeholder text when finished. Be sure to:

Indicate which changes you require for acceptance versus which changes you recommendAddress any conflicts between the reviews so that it's clear which advice the authors should followProvide specific feedback from your evaluation of the manuscript

Please ensure that your decision is justified on PLOS Global Public Health’s publication criteria and not, for example, on novelty or perceived impact.

Please submit your revised manuscript by . If you will need more time than this to complete your revisions, please reply to this message or contact the journal office at globalpubhealth@plos.org. Please include the following items when submitting your revised manuscript:

We look forward to receiving your revised manuscript.

Kind regards,

Laila Akbar Ladak, PhD, MScN, BScN, RN

Section Editor

Journal Requirements:

Additional Editor Comments (if provided):

Reviewers' comments:

Reviewer's Responses to Questions

**Comments to the Author**

1. Does this manuscript meet PLOS Global Public Health’s publication criteria? Is the manuscript technically sound, and do the data support the conclusions? The manuscript must describe methodologically and ethically rigorous research with conclusions that are appropriately drawn based on the data presented.

Reviewer #1: Yes

Reviewer #2: Yes

2. Has the statistical analysis been performed appropriately and rigorously?

Reviewer #1: N/A

Reviewer #2: N/A

3. Have the authors made all data underlying the findings in their manuscript fully available (please refer to the Data Availability Statement at the start of the manuscript PDF file)?

Reviewer #1: Yes

Reviewer #2: Yes

4. Is the manuscript presented in an intelligible fashion and written in standard English?

Reviewer #1: Yes

Reviewer #2: Yes

5. Review Comments to the Author

Reviewer #1: Overall this is a very good paper on an important topic of major global public health relevance. As such, it strongly deserves to be published as it can help inspire and inform:

a) Similar work in other settings

b) Follow-on research to address the many challenges identified here.

I have a few minor comments for authors to consider in order to further improve the value of their already strong, well described and well conducted study.

Title: Whilst I like the quote I wonder if dropping this and editing might help get more readers and more future citations. Title should ideally include: study type (qualitative); perhaps NOT mention study setting so as to broaden appeal (the messages in this paper for sure apply to many other settings, not just Malawi – see also https://www.natureindex.com/news-blog/studies-research-five-ways-increase-citation-counts

Abstract:

- Background might also be edited to make the point that the challenges faced in Malawi are in fact common to MANY other settings.

- Conclusions – might also mention clinical tools and approaches to help staff identify vulnerable infants who are at-risk of feeding/nutrition problems

Background

- Overall very good but could include some more references (see also last point re number of references cited). For instance:

o Be explicit about the mortality/morbidity numbers from suboptimal BF e.g. “Increasing breastfeeding to near-universal levels for infants and young children could save over 800000 children’s lives a year worldwide, equivalent to 13% of all deaths in children under two, and prevent an extra 20000 deaths from breast cancer every year.”https://www.thelancet.com/series/breastfeeding

https://www.evidentlycochrane.net/lancet-breastfeeding-series/

o Discuss the role of BF towards global SDG targets http://waba.org.my/archive/breastfeeding-a-key-to-sustainable-development-unicef-world-breastfeeding-week-2016-message/

o Give some definitions and a sentence or two extra re the differences between exclusive vs predominant vs partial BF: this is an important distinction and some readers might not be familiar with the issues. E.g. below is a good ref of the relative impact https://pubmed.ncbi.nlm.nih.gov/27013313/

- Were there some original aims and specific objectives of this study – the “purpose” is expressed but is vague

Methods

- Please say some more about the main study if this was really a secondary analysis. What were the aims/objectives of the wider ‘parent’ study and how do they differ from this one? Where are the results of the main study published?

Results:

- A table summarising the key characteristics of the respondents would be helpful (e.g. years of experience for the HCW; age and education level of the mothers)

- On page 9, authors make an important point that though information is provided, mother’s often forget/go back to old ways of doing things. Did they expand on what they thought were reasons for this? Was it for instance: the way info was provided (verbally/pictures/posters/what?) The quantity of info provided? Or just the fact that post-birth is such a busy and difficult time with so much going on?

On a related note, did you have any information on whether antenatal care/antenatal messages reduced this tendency to forget post-delivery counselling messages.

Please could authors also add a sentence and/or reference to say more about what the “counselling on best BF practices” actually involved. Was this part of general post-natal care guidance? If so, was this left up to each individual healthcare worker to decide what to do and say? Or are there locally tailored materials? Centre-specific guidelines? National Malawi Guidelines? International UNICEF or WHO or other guidelines?

- Page 10, Line 198. Can you say what an average time to discharge is after birth? IS there a minimum observation period before a mother/infant pair can be discharged? IS there any data on what the average stay in hospital post-delivery is in this setting?

- Page 11, line 227 – can you say what training the HCW might have had (or what support materials they might have) to assess adequacy of breastfeeding. Were any tools or guidelines used for this (e.g. see this SR for a list of available BF assessment tools https://pubmed.ncbi.nlm.nih.gov/33628437/ - perhaps some of these might be recommended in the discussion of this paper)

- Page 13, “Feasibility, challenges and local solutions” – in the discussion to accompany this section (rather than in the results section here), it would be helpful to note any data on BF indicators from the hospitals studied here. Are these similar to or higher/lower than national average. This would help readers understand how typical these challenges are.

Discussion

Please add a few lines on possible short / long term policy and practice implications e.g. wider use of breastfeeding assessment tools such as noted above; more training for staff; dedicated BF support staff; beginning messaging about BF in the antenatal period rather than perinatally when there’s too much going on for mothers to take all on-board.

References:

OK but a relatively small number. Consider this interesting paper https://www.nature.com/articles/news.2010.406 . It might both help future readers and citations of this paper to expand the number of references. What about, for example, mentioning some of the below:

- WHO Guidelines from here: https://www.who.int/health-topics/breastfeeding#tab=tab_1

- Other guidelines for providing support and managing problems for vulnerable infants e.g. https://www.ennonline.net/ife/iferesourcesoutputs
https://www.ennonline.net/mamicarepathway

https://www.cdc.gov/breastfeeding/resources/index.htm
https://www.globalbreastfeedingcollective.org/

Reviewer #2: General comments:

This is an interesting and informative article that sheds light on the “black-box” in health service delivery for breastfeeding support. The authors have done a nice job of capturing not only health worker perspectives, but that of the new mothers and other family members.

Minor comments:

Line 54: not sure if (2015) refers to a reference or the year, please make clear.

Reference 9 is missing journal information

Study design and Setting:

Please provide more description of the facilities i.e. BFHI certification, size of the catchment area, size of nursing staff/midwives etc. It may also be useful to include a table with these characteristics.

Recruitment and selection:

Although authors state purpose sampling, It seems as If there were two types of purposive sampling used one for the health workers and one for the mothers/family members. Can you please provide more clarity on the type of purposive sampling used i.e. convenience, criterion, used for each group?

How was it determined that a sample size of 5-10 participants was sufficient for data saturation, was this from formative work? If so please indicate

Data collection:

What was the rationale for including data from the pilot interviews? The questionnaires were adapted after piloting.

Results

Line 270-271 is repetitious of the quote in line 274. Could be more concise and allow the quote to make the point.

Lines 323-327 is repetitious of the quote in line 328. Could be more concise and allow the quote to make the point.

Discussion:

Please add to the discussion a paragraph explaining their results in the context of the Malawian health system and what are possible next steps e.g., are there are policies or program actions that government is undertaking to address staffing shortages? How about facility task shifting and/or community support for breastfeeding e.g., community health workers, peer networks etc.? Also breastfeeding support initiatives that include other family members.

Line 395-397: I think your argument here about generalizability to other facilities is weak as you’ve not adequately described these facilities.

6. PLOS authors have the option to publish the peer review history of their article (what does this mean?). If published, this will include your full peer review and any attached files.

**Do you want your identity to be public for this peer review?** For information about this choice, including consent withdrawal, please see our Privacy Policy.

Reviewer #1: No

Reviewer #2: No

---

## [Editor Report · Decision Letter 1]

29 Aug 2022

PGPH-D-22-00362R1

Beyond early initiation: A qualitative study on the challenges of hospital-based postpartum breastfeeding support

Dear Dr. Kinshella,

Thank you for submitting your manuscript to PLOS Global Public Health. After careful consideration, we feel that it has merit but does not fully meet PLOS Global Public Health’s publication criteria as it currently stands. Therefore, we invite you to submit a revised version of the manuscript that addresses the points raised during the review process.

Please address the following:

1.   There should be a table highlighting the participants characteristics. Also summarize the key information in the results section (e.g. years of experience for the HCW; age and education level of the mothers).

2.    COREQ guidelines document has been uploaded but needs to be filled in with the relevant information.

3. Some of the quotes are too long eg 278-299 which could be shrunk.

We look forward to receiving your revised manuscript.

Kind regards,

Laila Akbar Ladak, PhD, MScN, BScN, RN

Section Editor
---

## [Editor Report · Decision Letter 2]

17 Oct 2022

Beyond early initiation: A qualitative study on the challenges of hospital-based postpartum breastfeeding support

PGPH-D-22-00362R2

Dear Ms Kinshella,

We are pleased to inform you that your manuscript 'Beyond early initiation: A qualitative study on the challenges of hospital-based postpartum breastfeeding support' has been provisionally accepted for publication in PLOS Global Public Health.

Best regards,

Laila Akbar Ladak, PhD, MScN, BScN, RN

Section Editor
